# Mechanical Properties of Selective Laser Sintering Pure Titanium and Ti-6Al-4V, and Its Anisotropy

**DOI:** 10.3390/ma13225081

**Published:** 2020-11-11

**Authors:** Yuu Harada, Yoshiki Ishida, Daisuke Miura, Satoru Watanabe, Harumi Aoki, Taira Miyasaka, Akikazu Shinya

**Affiliations:** 1Department of Dental Materials Science, School of Life Dentistry at Tokyo, The Nippon Dental University, 1-9-20 Fujimi, Chiyoda-ku, Tokyo 102-8159, Japan; y-harada_d2117005@tky.ndu.ac.jp (Y.H.); yishida@tky.ndu.ac.jp (Y.I.); daisuke@tky.ndu.ac.jp (D.M.); s-watanabe_d2117004@tky.ndu.ac.jp (S.W.); haruaoki@tky.ndu.ac.jp (H.A.); miyasaka@tky.ndu.ac.jp (T.M.); 2Department of Life Science Dentistry, School of Life Dentistry at Tokyo, The Nippon Dental University, 1-9-20 Fujimi, Chiyoda-ku, Tokyo 102-8159, Japan; 3Department of Prosthetic Dentistry and Biomaterials Science, Institute of Dentistry, University of Turku, Lemminkaisenkatu 2, 20520 Turku, Finland

**Keywords:** additive manufacturing, selective laser sintering, Ti-6Al-4V, pure titanium, CAD/CAM

## Abstract

Selective laser sintering (SLS) is being developed for dental applications. This study aimed to investigate the properties of Ti-6Al-4V and pure titanium specimens fabricated using the SLS process and compare them with casting specimens. Besides, the effect of the building direction on the properties of the SLS specimens was also investigated. Specimens were prepared by SLS using Ti-6Al-4V powder or pure titanium powder. Casting specimens were also prepared using Ti-6Al-4V alloys and pure titanium. The mechanical properties (tensile strength and elongation), physical properties (surface roughness, contact angle, and Vickers hardness); corrosion resistors (color difference and corrosion), and surface properties (chemical composition and surface observation) were examined. Both Ti-6Al-4V and pure titanium specimens produced using the SLS process had comparable or superior properties compared with casting specimens. In comparing the building directions, specimens fabricated horizontally to the printing platform showed the greatest tensile strength, and the surface roughness scanned in the horizontal direction to the platform showed the smallest. However, there was no significant effect on other properties. Thus, the SLS process with Ti-6Al-4V powder and pure titanium powder has great performance for the fabrication of dental prosthesis, and there is a possibility for it to take the place of conventional methods.

## 1. Introduction

In recent years, additive manufacturing (AM) has been widely used in various fields and has been developed as a new molding technique [1]. This technology allows three-dimensional models to be built by fusing the material layer by layer [2]. In dentistry, the milling method is commonly used to fabricate prostheses in computer-aided design (CAD)/computer-aided manufacturing (CAM) systems; however, it is challenging to manufacture complicated shape objects because of the limitations of the milling tools, such as their size and applied angle [3]. In contrast, there are no limitations of the tools in AM, allowing fabrication of objects with complicated shapes. Moreover, AM allows many models to be manufactured at the same time. Therefore, in dentistry, AM might be a more efficient method compared with milling. 

Several additive manufacturing processes are being applied in dentistry to manufacture casting patterns, provisional restorations, surgical guides, or dental models [1,4,5,6,7,8]; however, it was difficult to fabricate final restorations with metal using the AM process due to the limitation of materials for additive manufacturing. Selective laser sintering (SLS) and selective laser melting (SLM), a powder bed fusion process, are applied to manufacture metal dental prostheses, such as fixed partial dentures [9]. In these molding methods, a thin layer of metal powder is selectively sintered by a laser beam based on 3D data. They have the advantage of being able to produce metal dental prostheses much faster than the conventional casting method because they do not require wax-up and investment. 

Recently, titanium materials have been developed as powders for the AM process. There are several reports showing that frameworks and clasps of removable dentures and fixed dentures can be produced by AM using titanium materials [10,11,12,13,14,15,16]. Moreover, Iseri et al. [17] reported increased bonding strength of ceramics to titanium alloy specimens molded by AM compared to those of titanium alloy specimens prepared by casting. However, the mechanical properties or resistance to corrosion of AM-molded titanium specimens have not been clarified. 

Therefore, this study aimed to clarify the properties of specimens molded by AM using titanium alloy (Ti-6Al-4V) powder and pure titanium powder. In addition, we investigated the effect of the building direction on the properties of the specimens. Finally, we compared the properties of the AM specimens with those of specimens cast with titanium alloy and pure titanium to examine the applicability of AM with titanium materials.

## 2. Materials and Methods

### 2.1. Specimen Preparation

The metal materials used in this study are listed in Table 1. Ti-6Al-4V specimen was molded using Ti-6Al-4V alloy powder and SLS 3D printer (M2, CONCEPT LASER, Lichtenfels, Germany). Using pure titanium (hereafter pure Ti) powder, specimens were molded using an SLS 3D printer (EOS M280, EOS, Krailling, Germany). Dumbbell-shaped specimens (diameter of the parallel part: 2 × 35 mm (DU)) and block specimens (10 × 10 × 20 mm (BL)) were prepared with each material.

The DU was molded to set the angle formed by the building direction and major axis at 0°, 45°, and 90° (Figure 1). For the molding of BL, a 10 × 10 mm plane horizontal to the building stage and the building direction were designated as the XY plane and Z-axis, respectively. Accordingly, a 10 × 20-mm plane is an XZ plane or YZ plane, but it is presented as a Z plane because they do not have meaning regarding molding. For post-processing, sandblasting was performed both DU and BL using a sandblaster (Cycle blaster Jr., Daiei Dental, Osaka, japan) with 400 µm glass beads (Sand-rough, Daiei Dental, Osaka, Japan).

As comparative controls, specimens with the same size as above were prepared by casting using the metals shown in Table 1. The DU casting pattern was prepared using an acrylic bar and paraffin wax (Figure 2). The BL casting pattern was prepared as follows: equally sized resin blocks were prepared using a 3D printer, an impression was taken using a silicone rubber impression material (putty type: Silde fit putty type, Shofu, Kyoto, Japan, regular type: Dent silicone–V, Shofu, Kyoto, Japan), and paraffin wax was poured into the impression.

These casting patterns were invested using a titanium-exclusive investing material (speed titanium investment, Shofu, Kyoto, Japan). Casting was prepared using a titanium-exclusive casting machine (Ti Cascom, Denken, Kyoto, Japan). Ti-6Al-4V casting was performed at a mold temperature of 430 °C and casting temperature of 1540–1650 °C; pure Ti casting was performed in an argon atmosphere at a mold temperature of 430 °C and casting temperature of 1668 °C, respectively. Post-processing was performed in the same manner as the specimens produced by SLS process.

### 2.2. Tensile Test

In the tensile test, the tensile strength and the elongation to time of fracture (matching method) of the DU specimens were measured. This test was performed at a crosshead speed of 1.0 mm/min using a universal testing machine (Autograph DCS-10T, Shimadzu, Kyoto, Japan). The SLS-0, SLS-45, SLS-90, and casting specimens were subjected to this test, and the influence of the building direction of the specimens on the mechanical properties was investigated.

### 2.3. Physical Property Test

The surface roughness, contact angle to water, and Vickers hardness were measured in a physical property test using BL specimens. Surface roughness was measured using a surface roughness measuring instrument (SURFCOM, Tokyo Seimitsu, Tokyo, Japan). The contact angle was measured with a contact angle meter (LSE-ME3, Nick, Saitama, Japan). The Vickers hardness was measured with a Vickers hardness tester (AVK-15, Akashi, Tokyo, Japan) at a measurement load of 50 kgf and a load holding time of 15 s. These measurements were performed on the surface condition as sintered. Next, the specimen surface was polished to a mirror-like finish, and the same measurements were performed on the surface after polishing. The properties of the specimen surface before (as sintered) and after polishing were compared. These tests were performed on the Z plane (SLS-Z), XY plane (SLS-XY), and the cast specimen, and the effect of the building direction of the specimen on the physical properties was investigated.

### 2.4. Immersion Test

Two types of immersion tests were performed: a discoloration test (using sodium sulfide and lactic acid solutions) and a corrosion test (using lactic acid solution). For these tests, the BL was cut into a plate of 10 × 10 × 1 mm using a precision cutting machine (IsoMet 1000, BUEHLER, Lake Bluff, IL, USA). The specimen cut vertical to the building stage (horizontal to the building direction) is referred to as SLS-Z, and the specimen cut horizontal to the building stage (vertical to the building direction) is referred to as SLS-XY. The two immersion tests were performed on SLS-Z, SLS-XY, and cast specimens, and the effects of the building direction of the specimen on discoloration and corrosion were investigated.

The plate specimen surface was polished with water-resistant abrasive paper (SiC abrasive paper, Struers, Tokyo, Japan (#400 to #1200)). Subsequently, the color, surface area, and weight of the specimen were measured (surface area and weight were measured only for the lactic acid immersed specimen). Color was measured with a colorimeter (CR-221, Konica Minolta, Tokyo, Japan). The surface area was measured with a digital microscope (VHX-2000, KEYENCE, Osaka, Japan). Weight was measured with a precision balance (MC210S, Sartorius, Goettingen, Germany). 

Sodium sulfide solution (50 mL, 0.1%) or lactic acid solution (10 g/L) was placed in a glass container (capacity approximately 150 mL). One specimen was immersed in the solution so that it did not contact the bottom of the container, and left at 37 °C for three (sodium sulfide solution) or seven days (lactic acid solution) according to ISO 10271:2011 [18]. After immersion, the same measurements as before immersion (color, surface area, and weight (surface area and weight were measured only for lactic acid immersed specimens)) were performed. 

In the discoloration test, the color difference (ΔE*ab) was calculated from chromaticity coordinates of the specimen measured before and after immersion. To relate the color change measured by the spectrophotometer to a clinical environment, the values of ΔE* ab were converted to National Bureau of Standards units (NBS units) through the equation, NBS units = ΔE*ab × 0.92.

In the corrosion test, weight loss due to corrosion (lactic acid weight loss) was calculated from the weight and surface area. Lactic acid weight loss was calculated using Formula (1): (1)$$\mathrm{Lactic acid weight loss}\;(\mu g/{\mathrm{cm}}^{2})=\frac{\mathrm{Weight before immersion}\;(\mu g)-\mathrm{Weight after immersion}\;(\mu g)}{\mathrm{Specimen}\;\mathrm{surface}\;\mathrm{area}\;({\mathrm{cm}}^{2})}$$

### 2.5. Surface Property Test

Chemical composition analysis and observation of metallographic structures were performed using plate specimens prepared in the same manner as the preparation of the specimens for the immersion tests. Chemical composition analysis was performed by fluorescent X-ray analysis (XRF) and X-ray diffraction (XRD). XRF was performed with a fluorescent X-ray analyzer (DELTA Professional, OLYMPUS, Tokyo, Japan). XRD was performed using a desktop X-ray diffractometer (MiniFlex, Rigaku, Tokyo, Japan). Surface structure was observed using a digital microscope (VHX-2000, KEYENCE, Osaka, Japan). The observation was performed on three patterns: as sintered, after polishing, and after etching (acidic solution (nitric acid:hydrofluoric acid:water = 4 1:5) was used for etching).

### 2.6. Test Repetitions and Statistical Processing

Each test was repeated six times from specimen preparation to measurement (n = 6). Two-way ANOVA was performed to analyze the results of the tensile test using DU (factor A: metal type (Ti-6Al-4V or pure Ti); factor B: building angle (SLS-0, SLS-45, SLS-90, or cast)). Three-way ANOVA was performed to analyze the results of the physical property tests using BL (factor A: metal type (Ti-6Al-4V or pure Ti); factor B: molding method (SLS-Z, SLS-XY, or cast); factor C: surface condition (as sintered or after polishing)). Two-way ANOVA was performed to analyze the results of the immersion test (factor A: metal type (Ti-6Al-4V or pure Ti); factor B: molding method (SLS-Z, SLS-XY, or cast)). One-way ANOVA was performed on XRF for each detected element. From the ANOVA results, Tukey’s multiple comparison was performed when there was a significant difference for the measurement item.

## 3. Results

### 3.1. Tensile Test

Table 2 shows the results of tensile test. A two-way ANOVA based on metal type (A) and building angle (B) was performed on the tensile strength test results. There were highly significant differences (*p* < 0.01) only in the main effects (A, B). Subsequent Tukey’s multiple comparison test showed there was a highly significant difference (*p* < 0.01) in metal type (A) (Ti-6Al-4V: 1047.85 MPa; pure Ti: 458.30 MPa; 95% confidence interval: 15.55), but no significant difference (*p* > 0.05) in building angle (B). Table 2 shows the results of Tukey’s multiple comparison for each metal type. For Ti-6Al-4V, the tensile strength of the SLS-90 specimen was significantly larger (*p* < 0.05) than that of the SLS-0 specimen. The tensile strength of the specimens molded by SLS was significantly larger (*p* < 0.05) than that of the cast specimens at any building angle. For pure Ti, the tensile strength of the SLS-90 specimen was significantly larger (*p* < 0.01) than that of the SLS-0 specimen. Comparing the tensile strengths of the specimens molded by SLS with those of the cast specimens, the tensile strength of SLS-45 and SLS-90 specimens was significantly larger (*p* < 0.01) than that of the cast specimen.

Two-way ANOVA was performed on the elongation results, and revealed highly significant differences (*p* < 0.01) only in main effects (A, B). Subsequent Tukey’s multiple comparison test revealed there was a highly significant difference (*p* < 0.01) in metal type (A) (Ti-6Al-4V: 12.13%; pure Ti: 18.03%; 95% confidence interval: 0.66), but no significant difference (*p* > 0.05) in building angle (B). Table 2 shows the results of Tukey’s multiple comparison for each metal type. There was no significant difference (*p* > 0.05) in elongation with regards to the building angle of SLS specimens. The elongation of the specimens molded by SLS was significantly smaller (*p* < 0.05) than that of the cast specimen at all building angles. For pure Ti, there was no significant difference (*p* > 0.05) between the molding methods or among the building directions.

### 3.2. Physical Property Test

Table 3 shows the results of physical property test. A three-way ANOVA based on metal type (A), molding method (B), and surface condition (C) was performed to analyze their effect on surface roughness, contact angle, and Vickers hardness. Regarding surface roughness there were highly significant differences (*p* < 0.01) in the main effects B, C, and in the interaction B × C. Figure 3 shows a graph of B × C and the results of Tukey’s multiple comparison. As shown in Figure 3, the sintered surface roughness was in the descending order of cast > SLS-Z > SLS-XY. After polishing, these values were greatly reduced for all the molding methods, and there was no significant difference between the molding methods or among the building directions (*p* > 0.05).

Regarding contact angle, there were highly significant differences (*p* < 0.01) in A, C, A × B, A × C, and A × B × C; and a significant difference (*p* < 0.05) in B. Figure 4 shows a graph of A × B × C and the results of Tukey’s multiple comparison. As shown in Figure 4, for Ti-6Al-4V, the SLS-Z and SLS-XY sintered contact angles were significantly larger (*p* <0.01) than that of the cast specimen. After polishing, these values were similar to those of the cast specimen before polishing. For pure Ti, the SLS-Z sintered contact angle was significantly smaller (*p* < 0.05) than that of the cast specimen. After polishing, this value was similar to that of the Ti-6Al-4V cast specimen before polishing.

Regarding Vickers hardness, there was a significant difference (*p* < 0.05) in A × C, and a highly significant difference (*p* < 0.01) in all the other main effects and interactions. Figure 5 shows a graph of A × B × C and the results of Tukey’s multiple comparison. As shown in Figure 5, for Ti-6Al-4V, the sintered Vickers hardness of the SLS-Z and SLS-XY specimens was significantly lower (*p* < 0.01) than that of the cast specimen. After polishing, the Vickers hardness of the cast specimen was similar to those of the SLS-Z and SLS-XY specimens, and there was no significant difference between the molding methods or among the building directions (*p* > 0.05). The Vickers hardness of pure Ti was significantly smaller than that of Ti-6Al-4V in both SLS and casting method (*p* < 0.05).

### 3.3. Immersion Test

Table 3 shows the results of immersion test.

Two-way ANOVA based on metal type (A) and molding method (B) was performed to assess their effect on the color difference (NBS unit) after immersion in either sodium sulfide or lactic acid solution, and lactic acid weight loss. Regarding the color difference after immersion in sodium sulfide solution, there was a highly significant difference (*p* < 0.01) in main effect A, but there was no significant difference (*p* > 0.05) in the molding method. The color difference of pure Ti was significantly increased (*p* < 0.01) compared with that of Ti-6Al-4V (Ti-6Al-4V: 12.64; pure Ti: 17.95; 95% confidence interval: 1.85).

Regarding lactic acid weight loss, there was a highly significant difference (*p* < 0.01) in main effect A, but there was no significant difference (*p* > 0.05) in molding method. The lactic acid weight loss of Ti-6Al-4V was significantly (*p* < 0.01) larger than that of pure Ti (Ti-6Al-4V: 137.96 µg/cm^2^; pure Ti: 56.68 µg/cm^2^; 95% confidence interval: 38.82).

Regarding the color difference after immersion in lactic acid solution, there was a highly significant difference (*p* < 0.01) in main effect A, and a significant difference (*p* < 0.05) in the interaction A × B. Figure 6 shows a graph of A × B and the results of Tukey’s multiple comparison. As shown in Figure 6, for Ti-6Al-4V, the color differences of the SLS-Z and SLS-XY specimens were significantly (*p* < 0.05) larger than that of the cast specimen. For pure Ti, there was no significant difference between the molding methods or among the building directions (*p* > 0.05).

### 3.4. Surface Property Test

Table 4 shows the results of the XRF. For Ti-6Al-4V, the titanium (Ti) fraction of the SLS-molded specimens was higher than that of the cast specimen. In particular, the Ti fraction of SLS-Z was significantly (*p* < 0.05) higher than that of the cast specimen. The vanadium (V) fraction of the SLS-molded specimens was significantly (*p* < 0.05) lower than that of the cast specimen. The iron (Fe) fraction was in descending order of SLS-XY > SLS-Z > cast. The aluminum (Al) and chromium (Cr) fractions did not significantly differ (*p* > 0.05) between specimens of the different molding methods. For pure Ti, the Ti fraction was in descending order of SLS-Z > SLS-XY > cast. The Fe fraction of SLS-Z specimens was significantly lower (*p* < 0.01) than that of the cast specimen. For the zirconium (Zr) fraction, there was no significant difference between the molding methods or among the building directions (*p* > 0.05).

Figure 7 and Figure 8 show the typical XRD spectra of the Ti-6Al-4V and pure Ti specimens, respectively. The peak intensities of XRD differed in each spectrum. Regarding the diffraction angles of the peaks, there were no differences in molding method for each metal. 

Figure 9 and Figure 10 show typical surface structure images of the Ti-6Al-4V and pure Ti specimens, respectively. For Ti-6Al-4V, the molding method did not appear to affect the surface structure pattern. For pure Ti, the as sintered images of the SLS specimens comprised a large block. In contrast, the cast specimen had a finer surface structure. After polishing and etching, they became smooth surfaces regardless of the molding method.

## 4. Discussion

In this study, several properties of the Ti-6Al-4V and pure Ti specimens produced using the SLS process were evaluated and compared to that of the casting specimens. Consequently, it was found that the specimens produced by SLS had comparable or superior properties compared with casting specimens. Besides, the building direction affected the tensile strength and surface roughness, but did not significantly affect other properties.

SLS is considered a promising AM method for manufacturing metal dental prostheses. To date, Co-Cr alloy powder has been primarily used for SLS in dentistry. It is difficult to manufacture fine titanium powders, and the production process to produce AM-qualified raw material were technically complicated [19]. For these reasons, SLS molding using titanium alloy powder and pure Ti powder has been delayed in practice. However, it has recently become possible to manufacture these powders for SLS [20,21,22]; therefore, SLS molding using these metal powders is promising for dentistry. The purpose of this study was to clarify the basic properties of molded objects manufactured by SLS using these metal powders.

### 4.1. Tensile Test

The tensile strength of Ti-6Al-4V was more than twice that of pure Ti. Moreover, the tensile strength values obtained for Ti-6Al-4V in this study satisfied those specified in JIS H4650 [23]. For pure Ti, the tensile strength of the SLS-molded specimens was in agreement with that of pure Ti (type 2 and type 3) specified in JIS H4650 [23]. The tensile strength of the cast specimen was in agreement with that of pure Ti (type 1 and type 2) specified in JIS H4650 [23]. Overall, the tensile strength values obtained in this study were similar to those previously reported for Ti-6Al-4V and pure Ti cast specimens [24,25]. The tensile strength of the SLS-90 specimen was greater than that of SLS-0 for all metal types. When the SLS building direction is the same as the long axis of the specimen, as in SLS-0, the layers are molded in the same direction as the direction of tensile stress. Therefore, tensile stress acts to separate the layers. In contrast, since the stress is vertical to the building direction in SLS-90 specimens, increasing their tensile strength. The tensile strengths of SLS-molded specimens were larger than those of the cast specimens for all metal types tested. This result was likely because of a difference in the composition of the metals used for SLS and casting. It is also possible the lack of casting defects in the SLS-molded specimens contributed to this result.

The elongation of pure Ti was larger than that of Ti-6Al-4V. The elongation values obtained in this study were close to those specified in JIS H4650 [23] (Ti-6Al-4V: 10% or more; pure Ti (type 3): 18% or more). These values are similar to those previously reported for Ti-6Al-4V and pure Ti cast specimens [26,27]. The SLS building angle had no effect in Ti-6Al-4V specimens. In contrast to tensile strength, the elongation values of SLS-molded specimens were smaller than those of the cast specimen. This result could be because of the difference in composition of the metals used for SLS and casting. In support of this, we found that the Fe fraction was low in the cast specimen, and this slight difference in composition could have influenced elongation. In pure Ti, there was no significant difference between the molding methods or among the building directions. Thus, the elongation of the pure Ti specimens was not affected by either the molding method or the metal composition. 

### 4.2. Physical Property Test

The metal type did not affect the sintered surface roughness. The surface roughness of the cast specimen was the largest of all the metal types, and the surface roughness of the SLS-Z specimen was larger than that of SLS-XY specimen for all metal types. Thus, the surface roughness scanning in the building direction was larger than that scanning in the horizontal direction to a building platform. This result was consistent with the observation that the SLS-90 specimen had a high tensile strength. It is considered that this result was obtained because the SLS-Z specimen surface has building steps. This result was compared the surface structure images. There was no clear difference between the molding methods in the surface structure images of the Ti-6Al-4V specimens. However, the pure Ti SLS-molded specimens had a surface structure comprising large blocks. This indicated a tendency consistent with the surface roughness results. After polishing, there was no significant difference in molding methods and metal type. This was consistent with the surface structure imaging results of the specimens after polishing. 

In Ti-6Al-4V SLS-molded specimens, the sintered contact angles were 90° or more, whereas that of the cast specimen was approximately 69°. After polishing, the contact angles were similar to that of the cast specimen before polishing, regardless of the molding method. Generally, the relationship between surface roughness and contact angle is expressed by the Cassie–Baxter Formula (2) [28]:cos θw = f (rcos θ + 1) − 1(2)

In this formula, θ is the contact angle, r is the area ratio of rough surface to plane (r ≧ 1), θw is the contact angle on the rough surface, and f is the ratio of liquid droplets entering the concave surface (if f = 1, the entire surface gets wet). Therefore, if f = 1, the larger the surface roughness, the larger the apparent contact angle on a hydrophobic surface (θ > 90°), and the smaller the apparent contact angle on a hydrophilic surface (θ < 90°). Therefore, for Ti-6Al-4V, the sintered contact angle of the SLS-molded specimen was large because its surface roughness was large. However, although the sintered surface roughness of the cast specimen was larger than that of the SLS-molded specimen, the contact angle of the cast specimen was smaller than that of the SLS-molded specimen. This could be because the f value is small; because the surface roughness of the cast specimen was larger; the concave surface could not be filled with water droplets. As a result, the contact angle of the cast specimen was small. For pure Ti, the surface roughness of the cast specimen was larger than that of the SLS-Z specimen. However, before polishing, the contact angle of the cast specimen was larger than that of the SLS-Z specimen. This could be because the effect of the f value was small. After polishing, the contact angles were 55°–70° for both metals. There was no difference between the molding method and metal type.

For Ti-6Al-4V, before polishing, the Vickers hardness of the cast specimen (650.99 HV) was larger than that of the SLS-molded specimen. During casting, the metal reacted with the refractory and binder materials of the investment material to form a hardened layer containing impurities on the surface layer of the specimen. We consider that the Vickers hardness of the cast specimen was increased by this hardened layer. Polishing removed the hardened layer; therefore, after polishing, the Vickers hardness of the cast specimen was approximately 331–400 HV, which was hardness of the SLS-molded specimen before and after polishing. No significant difference was not observed between the molding methods or among the building directions. For pure Ti, before polishing, the Vickers hardness of the cast specimen (278.51 HV) was larger than that of the SLS-molded specimens. However, after polishing, it was 219.09 HV, which was not different from the values of the SLS-molded specimens before and after polishing.

### 4.3. Immersion Test

The results of the color difference after immersion in sodium sulfide solution did not differ between the molding methods. The value of Ti-6Al-4V was 12.64 (standard deviation: 5.04), which was slightly smaller than, but within the standard deviation of, that of 15.2 previously reported for Ti-6Al-4V [29]. Furthermore, the previously reported value represents the distance in the Commission Internationale de l’éclairage (CIE) chromaticity coordinates by ΔE*ab. Therefore, to convert it into the NBS unit used in this study, it was multiplied by a coefficient corresponding to glossiness (0.92 [30]), giving a value of 13.98, which was close to that obtained in this study. The value of pure Ti was 17.95 (standard deviation: 1.32), which is close to that of 17.9 previously reported for pure Ti [29]. Converting it to NBS units, it became 16.49. Nevertheless, considering the error, we considered that this value was close to the value obtained in this study.

The molding method did not affect lactic acid weight loss. The value of Ti-6Al-4V was larger than that of pure Ti (Ti-6Al-4V: 137.96 µg/cm^2^; pure Ti: 56.68 µg/cm^2^). According to a previous report, when Ti-6Al-4V or pure Ti were immersed in lactic acid solution, the amount of Ti dissolved was 10 µg/cm^2^ and 23 µg/cm^2^, respectively [31]. Therefore, the amount of Ti dissolved from pure Ti was larger than that from Ti-6Al-4V. However, the corrosion resistance of pure Ti is generally superior to that of Ti-6Al-4V. Therefore, it is not possible to discuss the lactic acid weight loss based on the amount of Ti dissolved alone. Furthermore, the previous study examined the amount of Ti element dissolved in immersion solution using inductively coupled plasma-atomic emission spectrometry [31]. Therefore, considering the dissolution of Al and V, the values obtained in this study were reasonable as the value of lactic acid weight loss, which was examined by weight change before and after immersion.

For Ti-6Al-4V, the color difference of the SLS-XY specimen after lactic acid immersion was larger than that of the cast specimen. According to the results of the XRF on Ti-6Al-4V specimens, a larger Fe fraction was found in SLS compared with that of casting specimens. It suggests that the difference in composition caused the color difference. A previous study has reported a ΔE*ab color difference of approximately 4 after Ti-6Al-4V immersion in lactic acid [31]. Converting this value into NBS units, it was close to the value of the cast specimen obtained in this study. We consider that the values of the SLS-molded specimens (SLS-Z: 7.24; SLS-XY: 8.23) were larger than this value because of the difference in their composition, as described above. For pure Ti, the values of the color difference were 0.59–0.97. There was no difference between the molding methods or among the building directions. Previous study reported a value of 4 for pure Ti [31], which was rather large. However, considering the difference in corrosion resistance between Ti-6Al-4V and pure Ti, we consider that the values obtained in this study are consistent and accurate.

### 4.4. Surface Property Test

For Ti-6Al-4V, the Ti fraction of the particles used for SLS was higher than that of the alloy used for casting. Since the Fe fraction of the particles used for SLS was higher than that of the alloy used for casting, there was a slight difference in mechanical strength; however, we consider that they have essentially the same composition. For pure Ti, there was a slight difference in the Ti fraction between the particles used for SLS and the metal used for casting; however, we consider that they had also have the same composition.

There were differences in each spectrum for the peak intensities of XRD. Regarding the diffraction angles of the peaks (2θ), there were no differences between the molding method for each alloy. The results of XRD were obtained using the plate-shaped specimens, not from the powder specimen, so the crystal orientation was not randomized. Therefore, the comparisons of the relative intensities of each peak were considered to be meaningless, so in this study, we focused only on the comparison on the positions of the peaks, namely the diffraction angles of the peaks (2θ). Consequently, it could be considered that the spectra of SLS specimens were not affected by the molding directions or molding methods in either Ti-6Al-4V or pure Ti. The diffraction angles of peaks obtained in this study were close to those previously reported in the literature [32,33]. 

For Ti-6Al-4V, there was no obvious difference in the surface structures of specimens made using the different molding methods. The pure Ti as sintered SLS specimen comprised a large block. In contrast, the cast specimen had a finer surface structure. However, these differences were limited to the surface layer. After polishing and etching, there were no differences in the surface structure between the molding methods.

From the above, considering the mechanical, physical, chemical, and surface properties, Ti-6Al-4V and pure Ti specimens molded by SLS had either the same or better properties than cast specimens of Ti-6Al-4V and pure Ti. Only tensile strength and surface roughness were affected by the building direction of these specimens. Therefore, it can be concluded that the SLS process with either Ti-6Al-4V alloy powder or pure Ti powder is useful for clinical dentistry.

The main limitation of this study is that the environment of the oral cavity was not considered. That should be more complex than the experimental environment, so the difference must affect these properties. Further study is needed to clarify them.

## 5. Conclusions

In this study we compared the mechanical, physical, chemical, and surface properties of SLS-molded specimens using Ti-6Al-4V alloy powder and pure Ti powder with those of cast specimens of Ti-6Al-4V and pure Ti. The effect of the difference in the building direction of these specimens on the properties was also investigated. As a result, the following conclusions were obtained: 

(1) In both metals, the tensile strength of specimens with the same stress direction and building direction was small. The values of the specimens molded by SLS (Ti-6Al-4V: 1074.73 MPa, Pure Ti: 484.46 MPa) were larger than those of the cast specimens (Ti-6Al-4V: 967.21 MPa, Pure Ti: 379.85 MPa) for both metals. 

(2) For Ti-6Al-4V, the elongation of the specimens molded by SLS (11.27%) was smaller than that of the cast specimen (14.72%). In pure Ti, both values were equivalent.

(3) In both metals, the surface roughness of the surface of the building direction of SLS-molded specimens (Ti-6Al-4V: 5.04 µmRa, Pure Ti: 5.91 µmRa) was larger than other building direction (Ti-6Al-4V: 3.65 µmRa, Pure Ti: 3.40 µmRa). The value was smaller than that of the cast specimens (Ti-6Al-4V: 8.75 µmRa, Pure Ti: 7.71 µmRa). After polishing, there was no difference in surface roughness between the molding methods in both metals.

(4) For Ti-6Al-4V, the sintered contact angles of the SLS-molded specimens (98.14°) were larger than those of the cast specimen (68.86°). In Ti-6Al-4V after polishing and pure Ti, there were no obvious differences in the molding method.

(5) For Ti-6Al-4V, the sintered Vickers hardness of the SLS-molded specimens (366.84) was less than that of the cast specimen (650.99). In Ti-6Al-4V after polishing and pure Ti, there were no differences in the molding method.

(6) In both metals, there were no differences between the molding methods regarding color difference after sodium sulfide immersion. The color difference of pure Ti was larger than that of Ti-6Al-4V.

(7) In both metals, there were no differences between the molding methods regarding lactic acid weight loss. Lactic acid weight loss in Ti-6Al-4V was larger than in pure Ti.

(8) For Ti-6Al-4V, the color difference of the SLS-molded specimens after lactic acid immersion was greater than that of the cast specimen. In pure Ti, the values were small, and there was no difference between the molding methods.

(9) In both metals, there were no obvious differences between the molding methods for XRF, XRD, and surface structure.

From the above, it is concluded that SLS molding using either Ti-6Al-4V alloy powder or pure Ti powder is an extremely excellent molding method for clinical dentistry.

## Figures and Tables

**Figure 1 materials-13-05081-f001:**
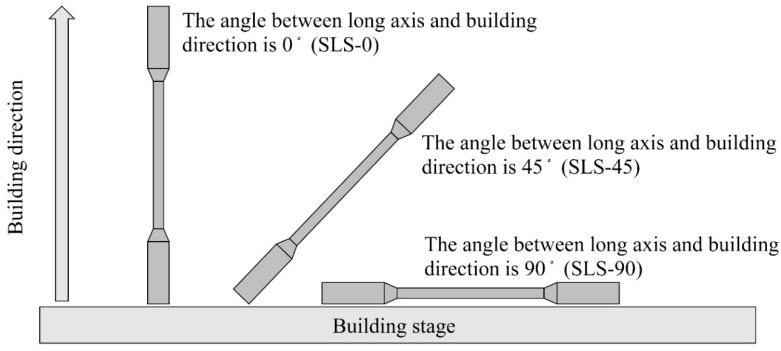
Building direction of SLS specimens.

**Figure 2 materials-13-05081-f002:**
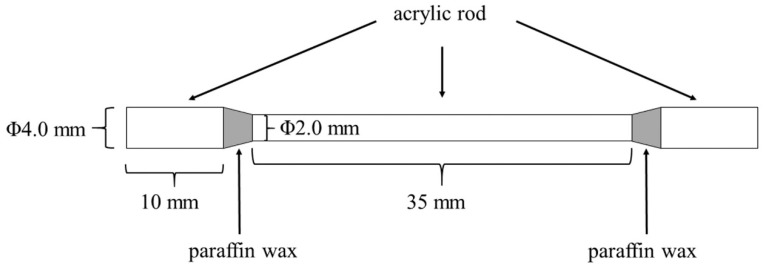
The pattern of the cast specimen.

**Figure 3 materials-13-05081-f003:**
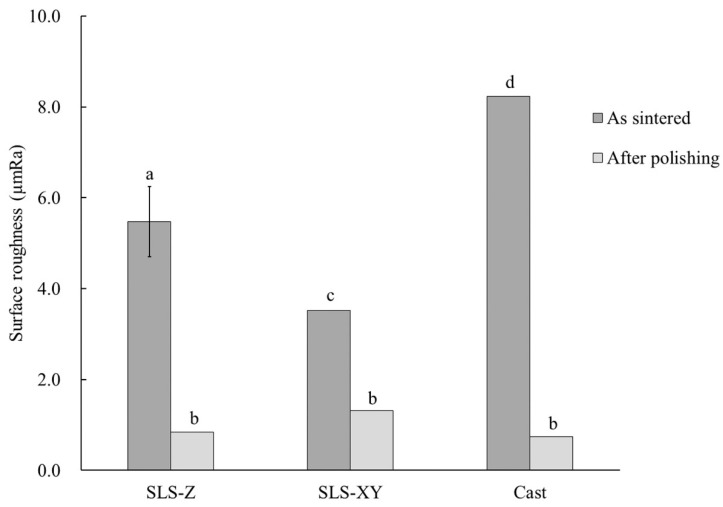
Surface roughness. The same lowercase letter indicates a combination with no significant difference (*p* > 0.05). Error bar represents 95% confidence interval.

**Figure 4 materials-13-05081-f004:**
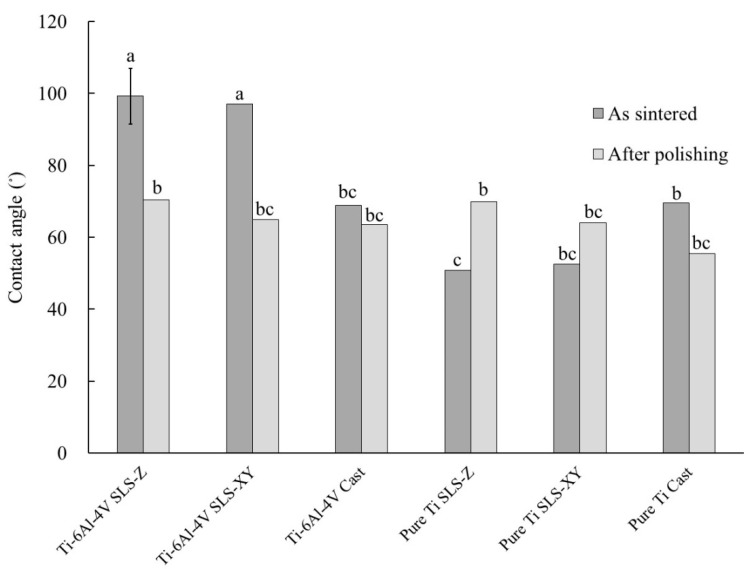
Contact angle. The same lowercase letter indicates a combination with no significant difference (*p* > 0.05). Error bar represents 95% confidence interval.

**Figure 5 materials-13-05081-f005:**
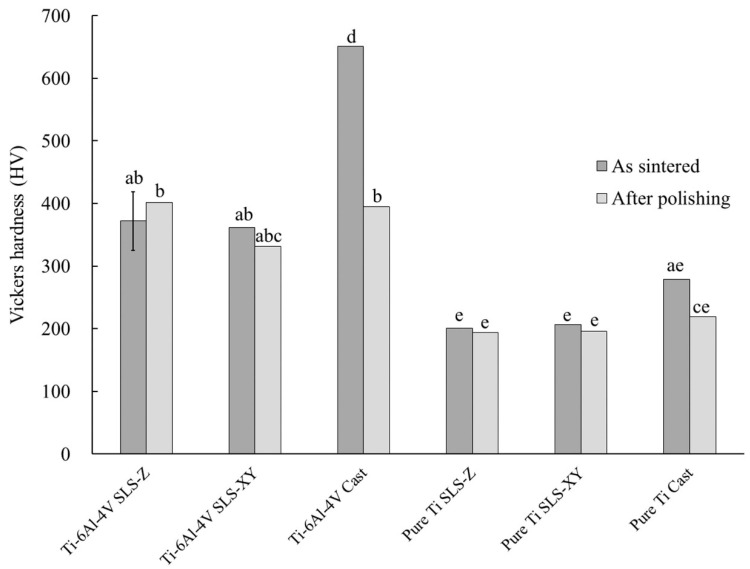
Vickers hardness. The same lowercase letter indicates a combination with no significant difference (*p* > 0.05). Error bar represents 95% confidence interval.

**Figure 6 materials-13-05081-f006:**
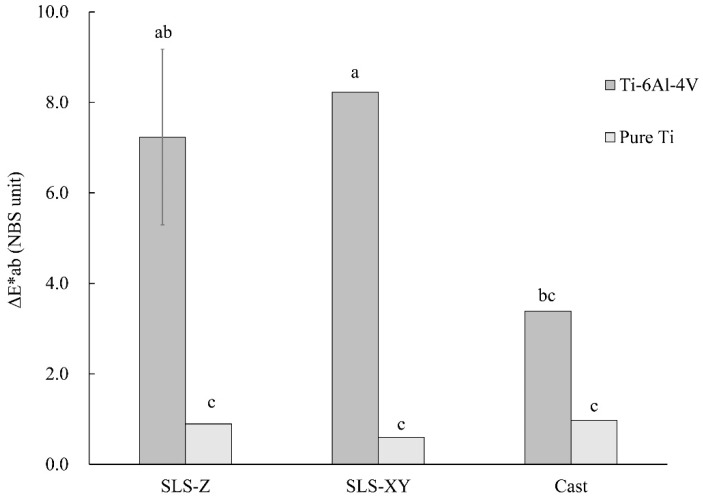
Color difference with lactic acid solution. The same lowercase letter indicates a combination with no significant difference (*p* > 0.05). Error bar represents 95% confidence interval.

**Figure 7 materials-13-05081-f007:**
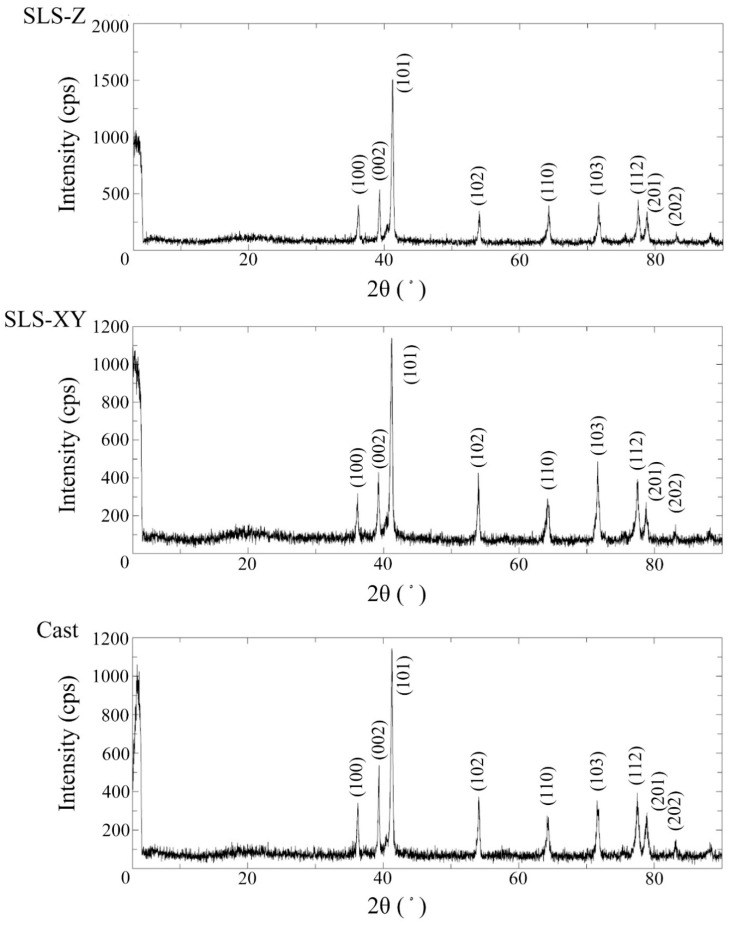
X-ray diffraction spectra of the SLS-Z, SLS-XY, and cast Ti-6Al-4V specimens.

**Figure 8 materials-13-05081-f008:**
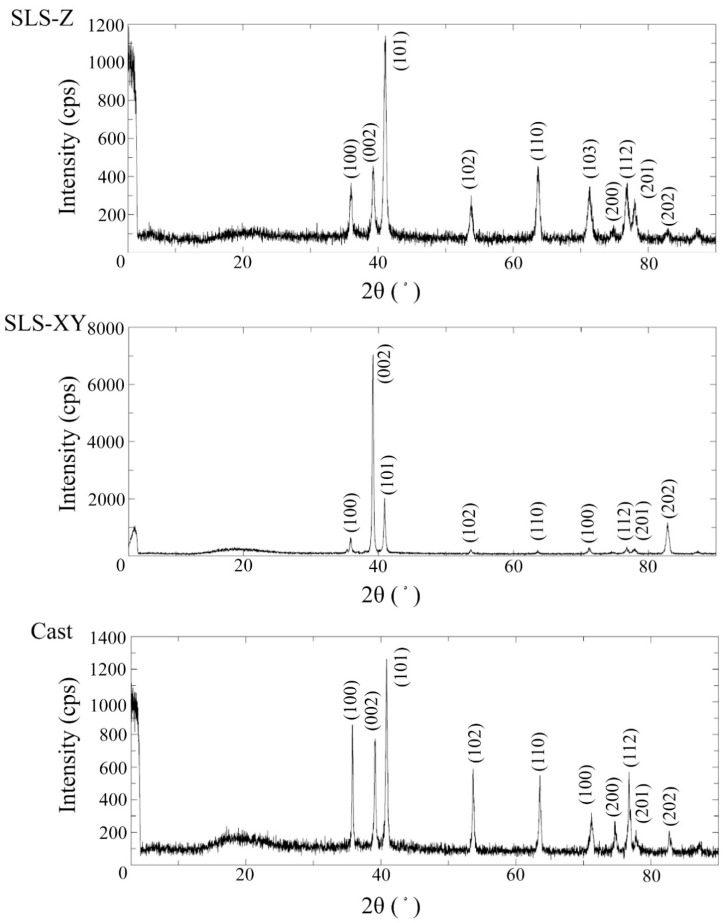
X-ray diffraction spectra of the SLS-Z, SLS-XY, and cast pure Ti specimens.

**Figure 9 materials-13-05081-f009:**
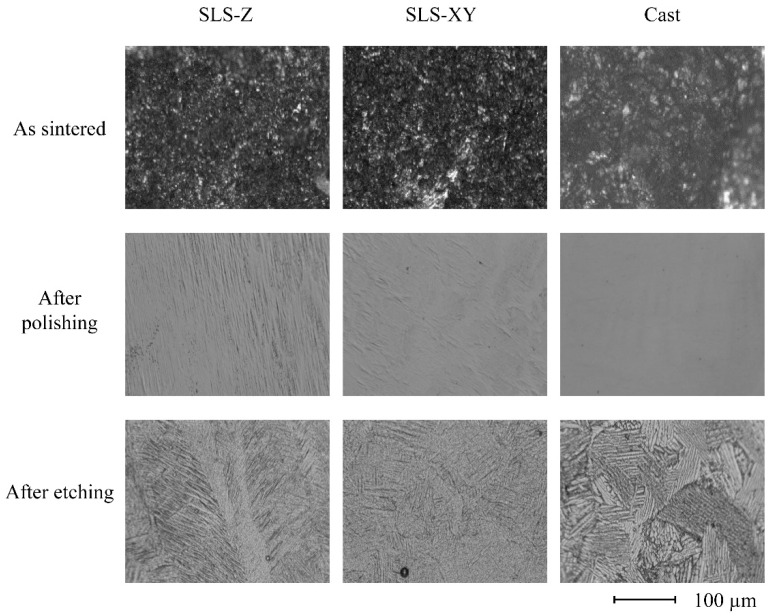
Digital microscope images of SLS-Z, SLS-XY, and cast Ti-6Al-4V specimens as sintered, after polishing, and after etching (×1000).

**Figure 10 materials-13-05081-f010:**
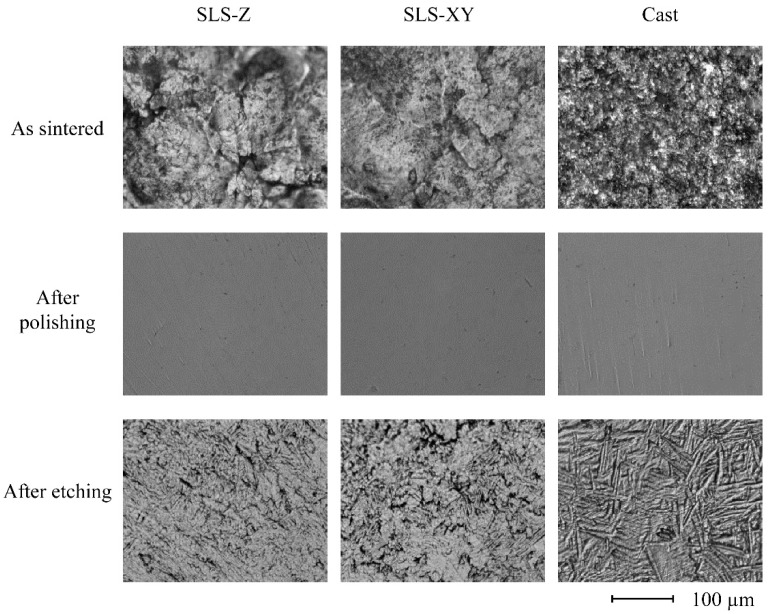
Digital microscope images of the SLS-Z, SLS-XY, and cast pure Ti specimens as sintered, after polishing, and after etching (×1000).

**Table 1 materials-13-05081-t001:** Metals used in this study.

Product Name	Constituent	Figure	Aspect (Particle Size)	Molding Method	Manufacturer
Rematitan CL	Ti-6Al-4V	powder	24 µm	SLS	Dentaurum (Ispringen, Germany)
Ti Alloy Gr.5	Ti-6Al-4V	rod	block	casting	Cosmo Metal (Gunma, Japan)
TILOP-45	Pure Ti	powder	38–45 µm	SLS	OSAKA Titanium Technologies (Hyogo, Japan)
Titan 100	Pure Ti	ingot	block	casting	Shofu (Kyoto, Japan)

**Table 2 materials-13-05081-t002:** Tensile strength and elongation.

Metal	Angle	Tensile Strength (MPa)			Elongation (%)		
Ti-6Al-4V	0	1044.12	(43.36)	A	10.90	(1.33)	A
	45	1061.89	(42.26)	A,B	11.56	(1.14)	A
	90	1118.17	(66.36)	B	11.34	(1.47)	A
	Cast	967.21	(38.37)	C	14.72	(1.67)	B
Pure Ti	0	430.51	(10.91)	a,b	17.62	(0.74)	a
	45	493.64	(16.69)	b,c	17.91	(0.09)	a
	90	529.22	(25.57)	c	17.82	(0.67)	a
	Cast	379.85	(31.85)	a	18.86	(3.37)	a

The same letter indicates a combination with no significant difference (*p* > 0.05). Standard deviations are given in parentheses.

**Table 3 materials-13-05081-t003:** Surface roughness, Contact angle, Vickers hardness, Lactic acid weight loss, and NBS in Na_2_S and lactic acid.

Metal	Surface	Surface Condition	Surface Roughness (µmRa)	Contact Angle (°)	Vickers Hardness (HV)	Lactic Acid Weight Loss (µg/cm^2^)	NBS Unit	
							Na_2_S	Lactic Acid
Ti-6Al-4V	SLS-Z	As sintered	5.04	99.22	371.82	-	-	-
			(0.24)	(6.37)	(30.83)			
		After polishing	0.91	70.46	401.06	58.99	13.10	7.24
			(0.36)	(6.63)	(138.78)	(39.83)	(5.15)	(4.45)
	SLS-XY	As sintered	3.65	97.05	361.85	-	-	-
			(1.46)	(8.72)	(19.04)			
		After polishing	1.21	64.87	331.50	153.86	13.29	8.23
			(0.27)	(8.61)	(56.34)	(121.42)	(7.05)	(3.31)
	Cast	As sintered	8.75	68.86	650.99	-	-	-
			(2.94)	(7.40)	(108.06)			
		After polishing	0.63	63.53	394.42	201.05	11.52	3.39
			(0.41)	(8.35)	(17.09)	(131.21)	(2.82)	(1.09)
Pure Ti	SLS-Z	As sintered	5.91	50.81	200.33	-	-	-
			(1.08)	(16.99)	(23.23)			
		After polishing	0.79	69.94	193.82	65.94	17.70	0.89
			(0.51)	(5.75)	(5.43)	(35.43)	(1.03)	(0.57)
	SLS-XY	As sintered	3.40	52.49	206.15	-	-	-
			(0.73)	(12.30)	(12.64)			
		After polishing	1.43	63.99	196.21	59.12	18.86	0.59
			(0.50)	(6.91)	(6.29)	(60.01)	(1.23)	(0.08)
	Cast	As sintered	7.71	69.49	278.51	-	-	-
			(2.92)	(7.83)	(49.71)			
		After polishing	0.86	55.43	219.09	44.97	17.28	0.97
			(0.51)	(11.34)	(28.78)	(25.01)	(1.31)	(0.24)
Standard deviations are given in parentheses.								

**Table 4 materials-13-05081-t004:** X-ray fluorescence analysis of elements (%).

Metal	Surface	Ti		Al		V		Fe		Cr		Zr	
Ti-6Al-4V	SLS-Z	89.74	A	6.00	A	4.06	A	0.19	A	0.00	A	-	
		(0.10)		(0.00)		(0.09)		(0.02)		(0.00)			
Ti-6Al-4V	SLS-XY	89.65	A,B	6.00	A	4.10	A	0.23	B	0.03	A	-	
		(0.15)		(0.00)		(0.14)		(0.03)		(0.06)			
Ti-6Al-4V	Cast	89.49	B	6.00	A	4.30	B	0.05	C	0.00	A	-	
		(0.11)		(0.00)		(0.14)		(0.02)		(0.00)			
Pure Ti	SLS-Z	99.99	a	-		-		0.01	a	-		0.00	a
		(0.01)						(0.01)				(0.00)	
Pure Ti	SLS-XY	99.97	a	-		-		0.03	a,b	-		0.00	a
		(0.02)						(0.02)				(0.00)	
Pure Ti	Cast	99.94	b	-		-		0.05	b	-		0.01	a
		(0.02)						(0.01)				(0.01)	

The same letter indicates a combination with no significant difference (*p* > 0.05). Standard deviations are given in parentheses. Ti: titanium, Al: aluminum, V: vanadium, Fe: iron, Cr: chromium, Zr: zirconium.

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
