# Peer review of "Mechanical Properties of Selective Laser Sintering Pure Titanium and Ti-6Al-4V, and Its Anisotropy"

_materials, 2020, doi:10.3390/ma13225081_

Round 1
Reviewer 1 Report
Manuscript ID: materials-988055
Title: Mechanical properties of selective laser sintering pure titanium and Ti-6Al-4V, and its anisotropy
Authors: Yuu HARADA et al.
The introduction is very extensive and contains a lot of valuable information. However, 2 paragraphs (line 60-68) are devoted to materials containing cobalt-chromium (Co-Cr) alloy powder. These alloys are not considered in this article. Does it worth paying much more attention to this alloys?
The methods and materials are described in detail with an indication of all the necessary materials, tests and equipment parameters.
line 144. Why is the experiment duration different for sodium sulfide and lactic acid?
Figures 7 and 8 are an insert from the software analysis results. The peaks on the XRD are not signed. The quality of the figures is low. XRD on Figure 7 are almost identical, however, there are significant differences in Figure 8. Authors do not write anything about the reasons for these differences in lines 265-267. What caused these differences?
Figures 9 and 10 do not have a scale bar.
lines 286-289. This text must be added to the Introduction chapter.
line 295. How does it expensive? Authors should provide more information (by $) or remove such statement.
Line 424. What is the chemical composition of the environment of the oral cavity? Why did the Authors not carry out tests simulating this oral environment impact?
There are no numbers and values in the conclusions of the article. Authors might reduce the discussion chapter and adding the main test values to the conclusions chapter.
Technical errors:
line 94. Figure 1 is not inserted into the article.
Author Response
The introduction is very extensive and contains a lot of valuable information. However, 2 paragraphs (line 60-68) are devoted to materials containing cobalt-chromium (Co-Cr) alloy powder. These alloys are not considered in this article. Does it worth paying much more attention to this alloys?
Response
Thank you very much for your expertise and time to review our manuscript. Your comments and suggestions will improve it. Our responses are detailed below, and the revisions in the manuscript were made accordingly.
The paragraphs regarding Co-Cr were removed from the introduction section.
The methods and materials are described in detail with an indication of all the necessary materials, tests and equipment parameters.
line 144. Why is the experiment duration different for sodium sulfide and lactic acid?
Response
The immersion tests were conducted according to ISO 10271 (Dentistry — Corrosion test methods for metallic materials), so the experimental durations were defined by it. The ISO number was added as a reference in the materials and methods section.
Figures 7 and 8 are an insert from the software analysis results. The peaks on the XRD are not signed. The quality of the figures is low. XRD on Figure 7 are almost identical, however, there are significant differences in Figure 8. Authors do not write anything about the reasons for these differences in lines 265-267. What caused these differences?
Response
The results of XRD were obtained using the plate-shaped specimens, not from the powder sample, so the crystal orientation was not randomized. Therefore, the comparisons of the relative intensities of each peak were considered to be meaningless, so in this study, we focused only on the comparison on the positions of the peaks, namely the diffraction angles of the peaks (2θ). Consequently, it could be considered that the spectra of SLS specimens were not affected by the molding directions or molding methods in either Ti-6Al-4V or pure Ti. We added the details mentioned above to the manuscript. Besides, we added the assignment of these spectra to Figure 7 and 8. The figures were replaced to them with high resolution.
Figures 9 and 10 do not have a scale bar.
Response
A scale bar was added in Figures 9 and 10.
lines 286-289. This text must be added to the Introduction chapter.
Response
The sentences were moved from the discussion section to the introduction section. Thank you for your valuable advice.
line 295. How does it expensive? Authors should provide more information (by $) or remove such statement.
Response
It was difficult to clarify the information on the price, so the statement was removed from the manuscript.
Line 424. What is the chemical composition of the environment of the oral cavity? Why did the Authors not carry out tests simulating this oral environment impact?
Response
There is a good report regarding the chemical composition of the environment of the oral cavity.
https://www.ncbi.nlm.nih.gov/pmc/articles/PMC2612946/
The environment of the oral cavity is very complex and differed among people, so it is not easy to replicate it. Besides, this study aimed to clarify the properties of specimens produced by the SLS process and investigate the advantages by comparing those of casting specimens. The effect of the oral cavity environment on their properties is under consideration for further study, so we hope that the results are gotten soon.
There are no numbers and values in the conclusions of the article. Authors might reduce the discussion chapter and adding the main test values to the conclusions chapter.
Response
The result values of the main tests were added in the conclusion section.
Technical errors:
line 94. Figure 1 is not inserted into the article.
Response
We apologize for it. Figure 1 was inserted into the manuscript.

Reviewer 2 Report
The authors report on mechanical and electrochemical properties of 3D printed Ti (and Ti compounds) and compare this to cast specimens. The experiments are performed with great care and analysis of the experimental results is done to the highest standards. Although the authors provide a certificate of English editing, there is still room for improvements in technical terminology. Furthermore, I would like to comment on the following in more detail:
Line 42ff: I do not think that the discussion of the classification provided by ISO 52900 is relevant for this work.
Line 53: Why was it difficult? No reason is given.
Line 83ff: How is it justified that raw materials from different sources were used? Why can the powders not be used for casting, too? Why were two different SLS machines used?
Line 94: The figure is missing.
Line 101f: Was there any mold release used? Was casting performed in air or under inert atmosphere?
Line 109ff: Was there any post-processing done to the parts after printing/casting, such as milling (to final dimension), sand blasting, etc?
Overall, I think that this work is highly interesting for material scientists in general as most results are generic enough to be used in all fields of engineering that utilize Ti. I thus strongly recommend publication after minor revisions.
Author Response
Comments for reviewer 2
The authors report on mechanical and electrochemical properties of 3D printed Ti (and Ti compounds) and compare this to cast specimens. The experiments are performed with great care and analysis of the experimental results is done to the highest standards. Although the authors provide a certificate of English editing, there is still room for improvements in technical terminology. Furthermore, I would like to comment on the following in more detail:
Response
Thank you very much for your expertise and time to review the manuscript. Your comments and advices will be improved our manuscript. Our responses are detailed below, and the revisions in the manuscript were made accordingly.
Line 42ff: I do not think that the discussion of the classification provided by ISO 52900 is relevant for this work.
Response
The sentences in the introduction section were removed and modified.
Line 53: Why was it difficult? No reason is given.
Response
For fabrication of dental prosthesis, metal and ceramics are widely used. However, there are no materials in additive manufacturing that can be applied to produce them. A sentence regarding that was added in the introduction section.
Line 83ff: How is it justified that raw materials from different sources were used? Why can the powders not be used for casting, too? Why were two different SLS machines used?
Response
In this study, casting specimens were produced by casting with a dental device that supports only for ingot, following the manufacture instruction. It is unclear whether casting with powder by the device has as good quality as casting with ingot. Thus, ingots were used for producing the casting specimens. The reason for using two different SLS machines is that the SLS machine for producing Ti-6Al-4V specimens does not support pure titanium so that another machine was needed for fabrication of them.
Line 94: The figure is missing.
Response
We apologize for it. Figure 1 was inserted into the manuscript.
Line 101f: Was there any mold release used? Was casting performed in air or under inert atmosphere?
Response
No molded release was used, and casting was performed under argon atmosphere. The materials and methods section was revised by adding some sentences.
Line 109ff: Was there any post-processing done to the parts after printing/casting, such as milling (to final dimension), sand blasting, etc?
Response
Sandblasting was performed as post-processing for both printing and casting specimens. Then, water-resistant abrasive papers were used to polish them to the final dimension. The sentences regarding that were inserted in the materials and methods section.
Overall, I think that this work is highly interesting for material scientists in general as most results are generic enough to be used in all fields of engineering that utilize Ti. I thus strongly recommend publication after minor revisions.
Response
Thank you again for the valuable comments. The manuscript was revised based on the comments, so please find it.

Round 2
Reviewer 1 Report
Authors have written a great commentary on the main points of the article.
The Introduction and Results sections have been improved. I wish the Author's success to continue their research in this direction in order to implement this development in dentistry.
This article can be published in Materials.